# Effects of a BMI1008 mixture on postoperative pain in a rat model of incisional pain

Geun Joo Choi[⊙], Eun Jin Ahn[⊙], Oh Haeng Lee[⊙], Hyun Kang[ID] *

Department of Anesthesiology and Pain Medicine, College of Medicine, Chung-Ang University, Seoul, Korea

⊙ These authors contributed equally to this work.
* roman00@naver.com

## Abstract

### Background

The purpose of this study was to evaluate the analgesic effect of BMI1008 (a new drug containing lidocaine, methylene blue, dexamethasone and vitamin B complex) and to investigate the analgesic effect of lidocaine and BMI-L (other components of BMI1008 except lidocaine) at different concentrations in a rat model of incisional pain.

### Methods

Male Sprague-Dawley rats (250–300 g) were used for the incisional pain model simulating postoperative pain. After the operation, normal saline, various concentrations of BMI1008, lidocaine with a fixed concentration of BMI-L, and BMI-L with a fixed concentration of lidocaine were injected at the incision site. The preventive analgesic effect was evaluated using BMI1008 administered 30 min before and immediately after the operation. In addition, BMI1008 was compared with positive controls using intraperitoneal ketorolac 30 mg/kg and fentanyl 0.5 μg/kg. The mechanical withdrawal threshold was measured with a von Frey filament.

### Results

The analgesic effect according to the concentration of BMI1008, lidocaine with a fixed concentration of BMI-L, and BMI-L with a fixed concentration of lidocaine showed a concentration-dependent response and statistically significant difference among the groups ($P < 0.001$, $P < 0.001$, and $P < 0.001$, respectively). The analgesic effect according to the time point of administration (before and after the operation) showed no evidence of a statistically significant difference between the groups ($P = 0.170$). Compared with the positive control groups, the results showed a statistically significant difference between the groups ($P = 0.024$).

### Conclusion

BMI1008 showed its analgesic effect in a rat model of incisional pain in a concentration-dependent manner. Moreover, BMI-L showed an additive effect on the analgesic effect of lidocaine.

**Data Availability Statement:** All relevant data are within the manuscript and its Supporting Information files.

**Funding:** HK: BMI Korea supported BMI 1008 materials and research grants for this research. No

role in the study design, data collection and analysis, decision to publish, or preparation of the manuscript. GJC: This research was supported by the Basic Science Research Program through the National Research Foundation of Korea (NRF) funded by the Ministry of Education, Science and Technology [Grant No. NRF-2020R1C1C1011263]. No role in the study design, data collection and analysis, decision to publish, or preparation of the manuscript.

**Competing interests:** BMI Korea supported BMI 1008 materials and research grants for this research. BMI Korea was only related to providing the products in development, but not related to employment, conultancy, patents and marketed products. This research was supported by the Basic Science Research Program through the National Research Foundation of Korea (NRF) funded by the Ministry of Education, Science and Technology [Grant No. NRF-2020R1C1C1011263]. BMI Korea and NRF had no role in the study design, data collection and analysis, decision to publish, or preparation of the manuscript. This did not alter our adherence to PLOS ONE policies on sharing data and materials.

## Introduction

The management of acute postoperative pain is an essential healthcare issue commonly faced by surgical patients. Although many developments have been made in terms of treatment, most patients undergoing surgery still experience considerable postoperative pain, and its mechanisms are complex [1]. Available treatment for postoperative pain is traditionally based on the use of opioids, which have limited use due to adverse effects, including respiratory depression, postoperative nausea and vomiting. Therefore, a number of strategies have been investigated to reduce opioid-related adverse effects in order to develop a promising and novel agent for more effective postoperative pain management.

The application of local anesthetics to surgical sites has been widely and increasingly used to reduce postoperative pain and the need for systemic analgesics because of its convenient application, safety, and low cost [2]. Further, it has the theoretical benefits of reducing the use of general anesthetics and their complications, hospital stays, and financial costs, and in increasing patient satisfaction. However, the use of multimodal agents in local injection and regional analgesia has not adopted broadly contrast to pursuing a multimodal approach in systemic administration of analgesics [3]. Even though each intervention has a low level of evidence, there are occasions where the therapeutic effect becomes apparent when the interventions are combined and applied [4]. Thus, the authors questioned whether local injection of mixture of analgesic agents would effect.

Lidocaine is one of the most commonly used local anesthetics because of its safety and fewer complications compared with other local anesthetics. However, the short duration of action of lidocaine (1–2 h without epinephrine) is an obstacle to its application for postoperative analgesia in clinical settings.

Methylene blue (MB) also has a theoretical benefit in terms of its analgesic effect because of its selective affinity for the nervous system [5]. Dexamethasone is a glucocorticoid that has been used to reduce inflammation and tissue damage in a variety of conditions, and its analgesic and anti-inflammatory effects after surgery have been explored [6]. Vitamin B complex and pH modified by $NaHCO_3$ also showed their pain-modulating effects [7, 8].

BMI1008 is a newly developed experimental drug that is a mixture of lidocaine, MB, dexamethasone, vitamin B complex, and $NaHCO_3$ (to adjust its pH). We hypothesized that BMI1008 would show analgesic effects and that BMI1008, lidocaine, and BMI-L (the other components of BMI1008 except lidocaine) would show a concentration-dependent response. Thus, the primary endpoint of this study was to evaluate the analgesic effect of BMI1008, and the secondary endpoint was to assess the concentration-dependent response of BMI1008, lidocaine, and BMI-L in a rat model of incisional pain. We also investigated the analgesic effect according to the time of BMI1008 administration and compared its analgesic effect with other analgesics, ketorolac, and fentanyl.

## Materials and methods

The experimental protocols were reviewed and approved by the Institutional Animal Care and Use Committee at Chung-ang University (2018-00092, 000943, 00095). All experiments were conducted according to the guidelines established by the National Institutes of Health, the policies of the International Association for the Study of Pain for the use of laboratory animals, and the recommended guideline in the Animal Research Reporting *In Vivo* Experiments (ARRIVE) statement [9].

## Preparation of study drug

BMI1008 (a mixture containing 1% lidocaine, MB 0.2 mg, dexamethasone 1 mg, vitamin B complex (including B1 0.062 mg, B2 0.012 mg, B6 0.062 mg, and B12 0.062 mg), and 5% $NaHCO_3$ per 1 mL as concentration of standard solution) was obtained from BMI Korea (Uiwang-si, Gyeonggi-do, Republic of Korea). The manufacturer provided experimental agents including BMI-L and BMI1008 concentrates. Ketorolac and fentanyl were purchased from Hanmi Pharmaceutical Corporation (Seoul, Korea) and Daewon Pharmaceutical Corporation (Seoul, Korea), respectively.

## Animal preparation

Adult male Sprague-Dawley rats weighing 250 to 300 g (Coretec Laboratories, Seoul, Korea) were used. The rats of 7 weeks old were acclimated in the colony room for a week before the experimental study and housed with two rats in each cage at $22 \pm 0.5°C$ with a 12:12-h light–dark cycle. Food and water were available *ad libitum*. Female rats were not used because hormonal fluctuations may affect the pain threshold [10]. Moreover, rats showing any abnormalities were excluded. On final day of experiment, rats were sacrificed in a carbon dioxide chamber ($CO_2$) two to four hours before dark-phase onset.

## Group allocation and study blindness

To assess the analgesic effect of BMI1008 and its components, the rats were randomly assigned to each group. Random assignment was based on a table generated by a computer applying Wei's Urn model using PASS$^{TM}$ 11 software (NCSS, Kaysville, UT, USA). The randomization code was generated by a statistician who was not otherwise involved in the study.

For allocation concealment, another investigator who was not involved in this study prepared syringes containing the study drugs for experiments. For intraplantar application, 1-ml syringes containing 0.2 mL of normal saline or study drugs were prepared, and for intraperitoneal (IP) injection, 5-ml syringes containing 2 ml of normal saline or study drugs were prepared. The syringes were covered with opaque tape and numbered sequentially according to a randomized list of respective experiments. Prepared syringes were delivered to a researcher in charge of the surgery. This researcher participated only in the surgery and was blinded to the group assignment.

## Surgical procedure

All surgical procedures were performed under sterile conditions. The rats received general anesthesia induced with 4% to 5% isoflurane in 100% oxygen (500 ml/min) inside a sealed clear plastic induction chamber until the rats became immobile. It was then maintained on a non-rebreathing anesthetic circuit nose cone using 1% to 2% isoflurane in 100% oxygen (200 ml/min) until the end of surgery to prevent the rats from suffering during the surgical procedure. Cefazolin (20 mg/kg; Chong Kun Dang Pharmaceutical Co., Korea) was administered subcutaneously prior to incision. The plantar surface of the left hind paw of each rat was prepared aseptically for surgery. The incisional pain model was created as previously reported [11], with minor modifications in the reported technique. In brief, at a point approximately 0.5 cm distal to the tibiotarsal joint on the plantar surface of the left hind paw, a 1-cm longitudinal skin incision extending toward the digits was made with a blade. The plantaris muscle was isolated, elevated slightly, and then incised longitudinally. Study drugs in prepared syringes were injected at the incision site (IS) over the prepared area. The incision was closed with two interrupted horizontal mattress sutures of 5–0 nylon. All rats were allowed to recover, and the sutures were removed on the third postoperative day.

### Experiment 1: Evaluation of concentration-dependent analgesic effect of BMI1008

The purpose of Experiment 1 was to evaluate the analgesic effect of BMI1008 according to the concentration of BMI1008. Sixty-six rats were randomly assigned to six groups as follows (n = 11 rats per group): Group Control, Group BMI X1, Group BMI X2, Group BMI X3, Group BMI X4, and Group BMI X5. The groups were given normal saline, BMI1008, double-concentrated BMI1008, triple-concentrated BMI1008, quadruple-concentrated BMI1008, and quintuple-concentrated BMI1008, respectively. The various concentrations of study drugs in the same volume (0.2 ml) were injected at the IS of the plantaris muscle before skin closure.

### Experiment 2: Evaluation of concentration-dependent analgesic effect of lidocaine with a fixed concentration of BMI-L

The purpose of Experiment 2 was to evaluate the analgesic effect according to the concentration of lidocaine, which is one of the components of BMI1008, while the concentration of BMI-L was fixed. Forty-eight rats were randomly assigned to four groups as follows (n = 12 rats per group): Group Control, Group 0.5-L, Group 1-L, and Group 2.5-L. The groups were given normal saline, lidocaine 0.5%+standard BMI-L, lidocaine 1%+standard BMI-L (BMI1008), and lidocaine 2.5%+standard BMI-L, respectively. The various concentrations of study drugs in the same volume (0.2 ml) were injected at the IS of the plantaris muscle before skin closure.

### Experiment 3: Evaluation of concentration-dependent analgesic effect of BMI-L with a fixed concentration of lidocaine

The purpose of Experiment 3 was to evaluate the analgesic effect according to the concentration of BMI-L while the concentration of lidocaine was fixed. Forty-eight rats were randomly assigned to four groups as follows (n = 12 rats per group): Group Control, Group BMI-L X0.5, Group BMI-L X1, and Group BMI-L X2. The groups were given normal saline, lidocaine 1% +half-concentrated BMI-L, lidocaine 1% +standard BMI-L (BMI1008), and lidocaine 1% + double-concentrated BMI-L, respectively. The various concentrations of study drugs in the same volume (0.2 ml) were injected at the IS of the plantaris muscle before skin closure.

### Experiment 4: Evaluation of analgesic effect according to the time point of administration: Pre-incisional versus post-incisional

The purpose of Experiment 4 was to evaluate the analgesic effect according to the time of BMI1008 administration. Sixty-eight rats were randomly assigned to two groups as follows (n = 34 rats per group): Group Before and Group After. Standard BMI1008 in the same volume (0.2 ml) was injected (1) at the intraplantar site 30 min before the operation in the Group Before and (2) at the IS of the plantaris muscle before skin closure in the Group After.

### Experiment 5: Comparison with positive control groups

The purpose of Experiment 5 was to assess the validity of the present study. Forty rats were randomly assigned to four groups as follows (n = 10 rats per group): Group Control, Group BMI, Group Ketorolac, and Group Fentanyl. The groups were given normal saline IP + normal saline IS, normal saline IP + BMI1008 IS, ketorolac 30 mg/kg IP + normal saline IS, and fentanyl 0.5 μg/kg IP + normal saline IS, respectively. The various concentrations of study drugs in the same volume (2 ml IP/0.2 ml IS) were injected at the peritoneum and IS of the plantaris muscle before skin closure.

## Behavioral measurements

Individual rats were placed on an elevated plastic mesh floor (8 x 8-mm perforations) under an overturned clear plastic cage (21 x 27 x 15 cm) and allowed to acclimate for 15 min. The rats were then tested to determine their withdrawal thresholds to mechanical stimuli using von Frey filaments (Stoelting Co., IL, USA). The filaments were applied vertically to the plantar aspect of the hind paw by administering pressure sufficient to gently bend the filaments. Filaments with bending forces of 4, 9, 20, 59, 78, 98, 147, and 254 mN were progressively applied until the hind paw was withdrawn or a bending force of 254 mN (the cutoff value) was reached. Each filament was applied three times at intervals of 3 min. The lowest bending force that caused paw withdrawal after application of the filament determined the mechanical withdrawal threshold (MWT) of the hind paw. The full lifting of the plantar surface off the mesh floor was considered a positive withdrawal response, and partial lifting, walking, hunching, stretching, or licking was not counted. After a response was observed, filaments with higher and lower bending forces were tested to confirm the MWT. The MWT assessment was carried out by one well-trained investigator who was unaware of the rats' group allocation. MWT was assessed according to the following schedule: at baseline (BL); 15 min after surgery (AS); 1, 2, 3, 4, 6, and 8 h AS; and 1, 2, 3, 5, and 7 d AS.

## Motor function tests

In order to identify the effect of BMI1008 on motor function, we used an accelerating Rota-Rod treadmill (Jeung-do Bio & Plant Co., Ltd., Seoul, Korea). Eighteen rats were randomly assigned to six groups as follows (n = 3 rats per group): Group Control, Group BMI X1, Group BMI X2, Group BMI X3, Group BMI X4, and Group BMI X5. The Rota-Rod test was performed 30 min after the intraplantar administration of study drugs. The rats were placed on the Rota-Rod treadmill, and its speed was gradually increased from 1 to 18 rotations per minute (rpm) for 120 s and maintained for another 30 s at 18 rpm [12]. The time at which the rat fell off the Rota-Rod was noted.

## Sample size calculation

The primary outcome measured was the withdrawal threshold to mechanical stimuli using von Frey filaments. To estimate the group size for the study, the MWT of the control group from a previous study was taken into account [13]. As the MWT of the Group Control did not pass the Shapiro-Wilk test, the data were transformed using the natural log. The averages of the natural log-transformed MWT (Ln(MWT)) at BL; 15 min AS; 1, 2, 3, 4, 6, and 8 h AS; and 1, 2, 3, 5, and 7 d AS were 4.28, 0.53, 1.87, 0.53, 0.88, 0.53, 0.41, 0.53, 0.83, 1.59, 2.10, 2.75, and 3.49 ln(mN), respectively. The standard deviations of the natural log-transformed MWT ranged from 0.56 to 2.1 ln(mN), and the autocorrelation between adjacent measurements on the same rat was 0.7. For our power calculations, we assumed that a first-order autocorrelation adequately represented the autocorrelation pattern. To compare between-group differences, we planned to use the Geisser-Greenhouse Corrected F test for the repeated-measures analysis of variance (ANOVA). We wanted to detect a 10%, 25%, 50%, 75%, and 100% increase in the MWT in the Group BMI X1, BMI X2, BMI X3, BMI X4, and BMI X5 compared with the Group Control in **Experiment 1**; a 50%, 75%, and 100% increase in the MWT in the Group 0.5-L, 1-L, and 2-L compared with the Group Control in **Experiment 2**; a 50%, 65%, and 80% increase in the MWT in the Group BMI-L X0.5, BMI-L X1, and BMI-L X2 compared with the Group Control in **Experiment 3**; no change and a 50% increase in the MWT in the Group Before and After in **Experiment 4**; and a 50%, 100%, and 100% increase in the MWT in the Group BMI, Ketorolac, and Fentanyl compared with the Group Control in **Experiment 5**.

With an α of 0.05 and a power of 80%, we needed 10 rats per group in **Experiment 1**, 15 rats per group in **Experiment 2**, 10 rats per group in **Experiment 3**, 31 rats per group in **Experiment 4**, and 9 rats per group in **Experiment 5**. A follow-up loss rate of 10% was applied. PASS 11™ software (NCSS) was used to calculate the sample size.

## Statistical analysis

The Shapiro-Wilk test was used to test the normality of variables. The times for falling from the Rota-Rod passed the normality test. Thus, the times for falling from the Rota-Rod were analyzed using an ANOVA followed by a Tukey post-hoc test. However, since the MWT did not pass the Shapiro-Wilk test, a natural log transformation of the MWT was performed, and the natural log-transformed MWT passed the Shapiro-Wilk test. We therefore assumed that the normal distribution assumption for the parametric test was not violated and decided to apply the Geisser-Greenhouse Corrected F test for the repeated-measures ANOVA.

Mauchly's sphericity test indicated that the assumption of sphericity had been violated in **Experiment 1** [($\chi^2$ (77) = 290.328, $P < 0.001$, Mauchly's W = 0.006)], **Experiment 2** [$\chi^2$ (77) = 258.33, $P < 0.001$, Mauchly's W = 0.002], **Experiment 3** [$\chi^2$ (77) = 327.52, $P < 0.001$, Mauchly's W = 0.003], **Experiment 4** [$\chi^2$ (77) = 862.87, $P < 0.001$, Mauchly's W<0.001], and **Experiment 5** [$\chi^2$ (77) = 201.18, $P < 0.001$, Mauchly's W = 0.002]. Thus, we used Wilks' lambda's multivariate analysis of variance (MANOVA) with each group as independent factors and Ln(MWT) at each time point (at BL; 15 min AS; 1, 2, 3, 4, 6, and 8 h AS; and 1, 2, 3, 5, and 7 d AS) as dependent variables. To compare the MWT at each time point, a univariate ANOVA or t-test with Bonferroni correction (α = 0.05/ 12 = 0.0042) was used. When the homoscedasticity requirement using Levene's test for homogeneity of variances was not met in the ANOVA, we used Welch's corrected ANOVA. Tukey's or Tamhane's T2 post-hoc test was used when ANOVA or Welch's corrected ANOVA was significant to identify the groups with statistically significant mean differences.

Additionally, between-group differences for Ln(MWT) were analyzed with a linear mixed-effects model (LMEM), which was made with times at BL; 15 min AS; 1, 2, 3, 4, 6, and 8 h AS; and 1, 2, 3, 5, and 7 d AS and groups as independent fixed factors and individual patients as random effects.

We also drew the response surface analysis plot of the drug combination to summate the analgesic effect of lidocaine and BMI-L [14]. The analgesic effect of each combination was calculated using means of the area under the curve (AUC) for 3 d in Ln(MWT). Moreover, seven points of drug combination were selected when BMI-L was less than 2x and the concentration of lidocaine was less than 2.5%.

Individual measurements were expressed as the mean ± standard error and analyzed with SPSS 23.0 (IBM Corp., Armonk, NY, USA). A P-value of less than 0.05 was considered statistically significant.

# Results

## Study animals

All of the rats completed the present study. Throughout the experimental period, the rats remained well groomed and appeared to ingest a normal amount of food and water. Except for impaired weight bearing on the area of the incision, gait appeared unaffected. None of the rats had complications of the surgical wound.

## Experiment 1: Evaluation of concentration-dependent analgesic effect of BMI1008

Fig 1 shows the changes in the Ln(MWT) measured at BL; 15 min AS; 1, 2, 3, 4, 6, and 8 h AS; and 1, 2, 3, 5, and 7d AS. The results of the MANOVA showed a statistically significant difference among the groups (F[65.00, 235.51] = 6.924, $P < 0.001$: Wilks' lambda = 0.007, partial $\eta^2$ = 0.636).

The values of Ln(MWT) at BL were not significantly different among the groups. Compared with the Group Control, the values of Ln(MWT) from 15 min AS to 5 d AS in the Group BMI X3, BMI X4, and BMI X5 and from 15 min AS to 3 d AS in the Group BMI X1 and BMI X2 were significantly increased. Compared with the Group BMI X1, a significant increase in Ln(MWT) was observed from 3 h AS to 1 d AS in the Group BMI X5 and from 3 h to 8 h AS in the Group BMI X4. Compared with the Group BMI X2, the values of Ln(MWT) at 6 h and 8 h AS in the Group BMI X5 and at 6 h AS in the Group BMI X4 were significantly increased.

The LMEM showed significant differences among the groups (F[5, 742.55] = 949.43, $P < 0.001$). Compared with the Group Control, there were significant differences of estimated difference in means (MD): 2.55 [2.41–2.70] in Group BMI X1, 2.76 [2.62–2.91] in Group BMI X2, 2.96 [2.81–3.10] in Group BMI X3, 3.23 [3.09–3.38] in Group BMI X4, and 3.33 [3.19–3.47] in Group BMI X5.

## Experiment 2: Evaluation of concentration-dependent analgesic effect of lidocaine with fixed concentration of BMI-L

The results of the MANOVA showed a statistically significant difference among the groups (F [39.00, 95.51] = 13.927, $P < 0.001$: Wilks' lambda = 0.004, partial $\eta^2$ = 0.847).

The values of Ln(MWT) at BL were not significantly different among the groups. Compared with the Group Control, the values of Ln(MWT) from 15 min AS to 3 d AS in the Group 1-L and 2.5-L and from 15 min AS to 1 d AS in the Group 0.5-L were significantly increased. Compared with the Group 0.5-L, a significant increase in Ln(MWT) was observed from 15 min AS to 3 d AS in the Group 1-L and 2.5-L. Compared with the Group 1-L, the values of Ln(MWT) from 4 to 8 h AS were significantly increased in the Group 2.5-L (Fig 2).

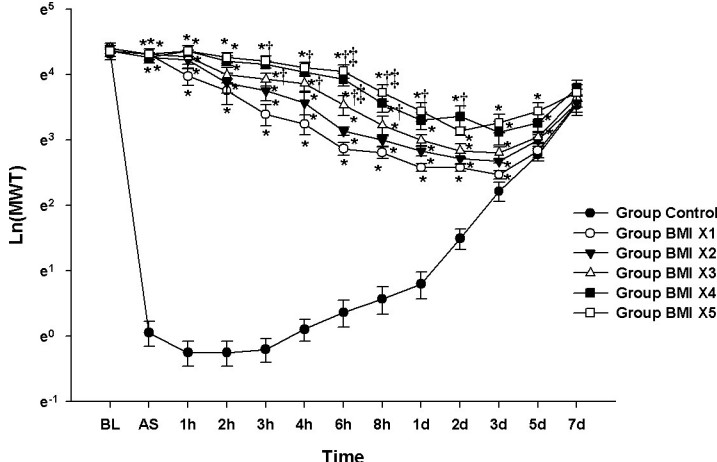

**Fig 1. Evaluation of analgesic effect according to dose of BMI1008: BMI1008 dose response test.** The graphs are presented as means ± standard error of the mean of the log-transformed mechanical withdrawal threshold with von Frey filaments. BL: baseline, AS: after surgery. * P <0.05 compared with Group Control, † P<0.05 compared with Group BMI X 1, ‡ P<0.05 compared with Group BMI X2.

The LMEM showed significant differences among the groups (F[3, 347.93] = 851.63, $P < 0.001$). Compared with the Group Control, there were significant differences of MD: 2.38 [2.23–2.54] in Group 0.5-L, 3.24 [3.09–3.40] in Group 1-L, and 3.65 [3.50–3.81] in Group 2.5-L.

## Experiment 3: Evaluation of concentration-dependent analgesic effect of BMI-L with fixed concentration of lidocaine

The results of the MANOVA showed a statistically significant difference among the groups (F [39.00, 142.89] = 15.935, $P < 0.001$: Wilks' lambda = 0.007, partial $\eta^2$ = 0.809).

The values of Ln(MWT) at BL and 15 min AS were not significantly different among the groups.

Compared with the Group Control, the values of Ln(MWT) from 15 min AS to 3 d AS in the Group BMI-L X0.5 and BMI-L X1 and from 15 min AS to 1 d AS in the Group BMI-L X0.5 were significantly increased. Compared with the Group BMI-L X0.5, a significant increase in Ln(MWT) was observed from 15 min AS to 1 h AS and from 3 h AS to 3 d AS in the Group BMI-L X2 and at 15 min AS and from 8 h AS to 3 d AS in the Group BMI-L X1 (**Fig 3**).

The LMEM showed significant differences among the groups (F[3, 560.01] = 698.75, $P < 0.001$). Compared with the Group Control, there were significant differences of MD: 2.17 [2.03–2.31] in Group BMI-L X0.5, 2.61[2.47–2.75] in Group BMI-L X1, and 2.88 [2.74–3.02] in Group BMI-L X2.

## Experiment 4: Evaluation of analgesic effect according to the time point of administration: Pre-incisional versus post-incisional

The results of the MANOVA showed no evidence of statistically significant differences among the groups (F[13.00, 54.00] = 1.444, $P = 0.170$: Wilks' lambda = 0.470, partial $\eta^2$ = 0.230). There was no evidence of differences among the groups at each time point (**Fig 4**).

The LMEM showed no evidence of significant differences among the groups (F[1, 358.18] = 0.989, $P = 0.321$).

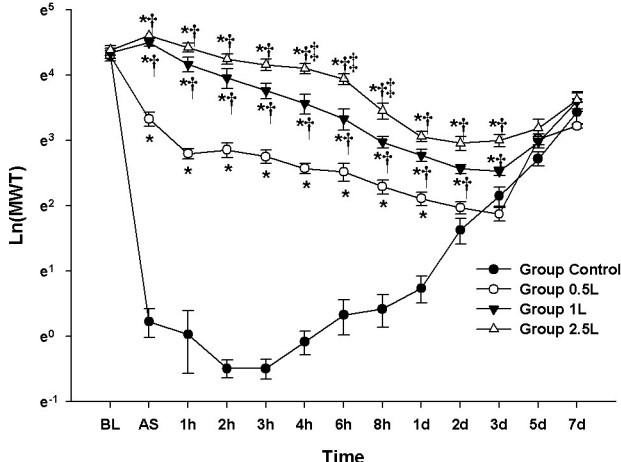

**Fig 2. Evaluation of analgesic effect according to dose of lidocaine: Lidocaine dose response test.** The graphs are presented as means (± standard error of the mean) of the log-transformed mechanical withdrawal threshold with von Frey filaments. BL: baseline, AS: after surgery. * P <0.05 compared with Group Control, † P<0.05 compared with Group Lidocaine 0.5% + Fixed BMI, ‡ P<0.05 compared with Group Lidocaine 1%+ Fixed BMI.

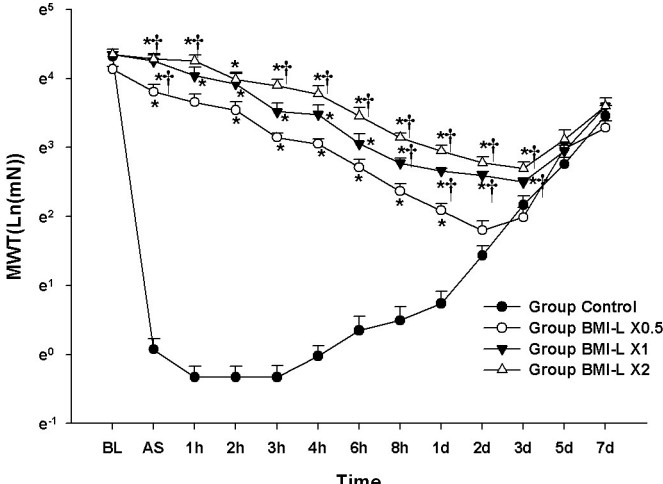

**Fig 3. Evaluation of analgesic effect according to dose of BMI component: BMI component dose response test.** The graphs are presented as means (± standard error of the mean) of the log-transformed mechanical withdrawal threshold with von Frey filaments. Mechanical withdrawal threshold with von Frey filaments. BL: baseline, AS: after surgery. * $P < 0.05$ compared with Group Control, † $P < 0.05$ compared with Group Fixed Lidocaine + BMI X 0.5.

## Experiment 5: Comparison with the positive control groups

The results of the MANOVA showed a statistically significant difference among the groups (F [39.00, 71.82] = 11.505, $P < 0.001$: Wilks' lambda = 0.003, partial $\eta^2$ = 0.858). Compared with the Group Control, the values of Ln(MWT) from 15 min AS to 3 d AS in the Group BMI, Ketorolac, and Fentanyl were significantly increased. There were statistically significant differences at 6 and 8 h AS between the Group BMI and Fentanyl (**Fig 5**).

The LMEM showed significant differences among the groups (F[3, 560.01] = 698.75, $P < 0.001$). Compared with the Group Control, there were significant differences of MD: 1.19 [1.03–1.36] in Group BMI, 1.02 [0.85–1.18] in Group Ketorolac, and 0.93 [0.77–1.10] in Group Fentanyl.

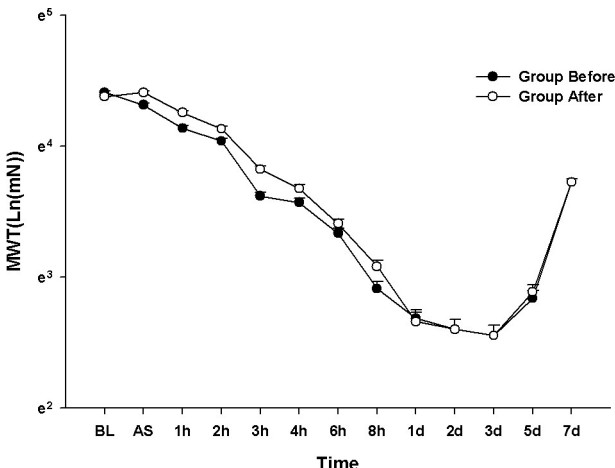

**Fig 4. Evaluation of analgesic effect according to time of administration: Before vs. after test.** The graphs are presented as means (± standard error of the mean) of the mechanical withdrawal threshold with von Frey filaments. Mechanical withdrawal threshold with von Frey filaments. BL: baseline, AS: after surgery.

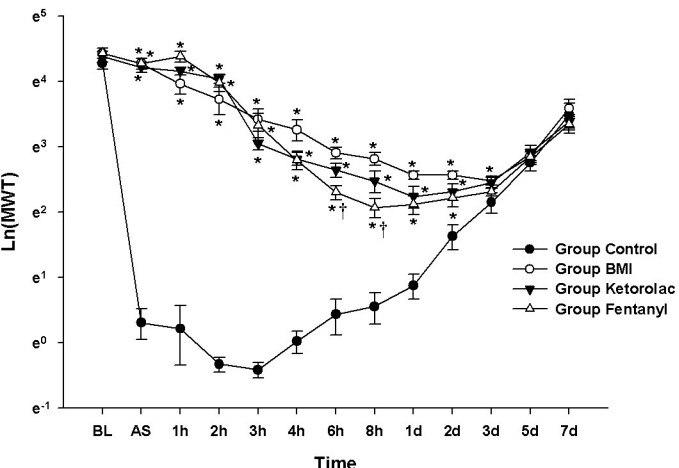

**Fig 5. Comparison with positive control groups.** The graphs are presented as means (± standard error of the mean) of the mechanical withdrawal threshold with von Frey filaments. Mechanical withdrawal threshold with von Frey filaments. BL: baseline, AS: after surgery. * P <0.05 compared with Group Control, † P<0.05 compared with Group BMI.

When the comparison except the Group Control was performed, the MANOVA and LMEM showed a statistically significant difference among the groups (F[26.00, 30.00] = 2.127, $P$ = 0.024: Wilks' lambda = 0.124, partial $\eta^2$ = 0.648 and F[2, 273.96] = 6.841, $P$ = 0.001, respectively).

## Motor function tests

The administration of BMI1008 showed no significant effect on motor function as measured by the Rota-Rod test 30 min after intraplantar administration of BMI1008 compared with the Group Control (p = 0.894). The times (means ± standard deviation, seconds) for falling from the Rota-Rod were 107 ± 15 in Group Control, 109 ± 11 in Group BMI X1, 107 ± 13 in Group BMI X2, 101 ± 10 in Group BMI X3, 102 ± 12 in Group BMI X4, and 104 ± 11 in Group BMI X5 (**Fig 6**).

## Response surface analysis

The AUC of Ln(MWT) showed an approximately linear positive correlation with BMI-L and lidocaine concentration. Moreover, co-administration of BMI-L and lidocaine augmented the AUC of Ln(MWT) compared with the administration of single drugs. Gross concavity of the plane was not observed, suggesting a higher possibility of the additivity assumption and a lower possibility of the antagonism (infra-additivity) assumption (**Fig 7**).

## Discussion

In this study, BMI1008 showed a significant analgesic effect in a rat model of incisional pain. Not only the concentration of BMI1008 but also the concentration of each component of BMI1008 (both a variable concentration of lidocaine while the concentration of BMI-L was fixed and a variable concentration of BMI-L while the concentration of lidocaine was fixed) showed a concentration-dependent response in the analgesic effect. However, the preventive administration of BMI1008 showed no significant difference compared to the administration of BMI1008 AS. Drug interaction between lidocaine and BMI-L were more likely to be additivity than antagonism (infra-additivity) when these drugs were co-administered.

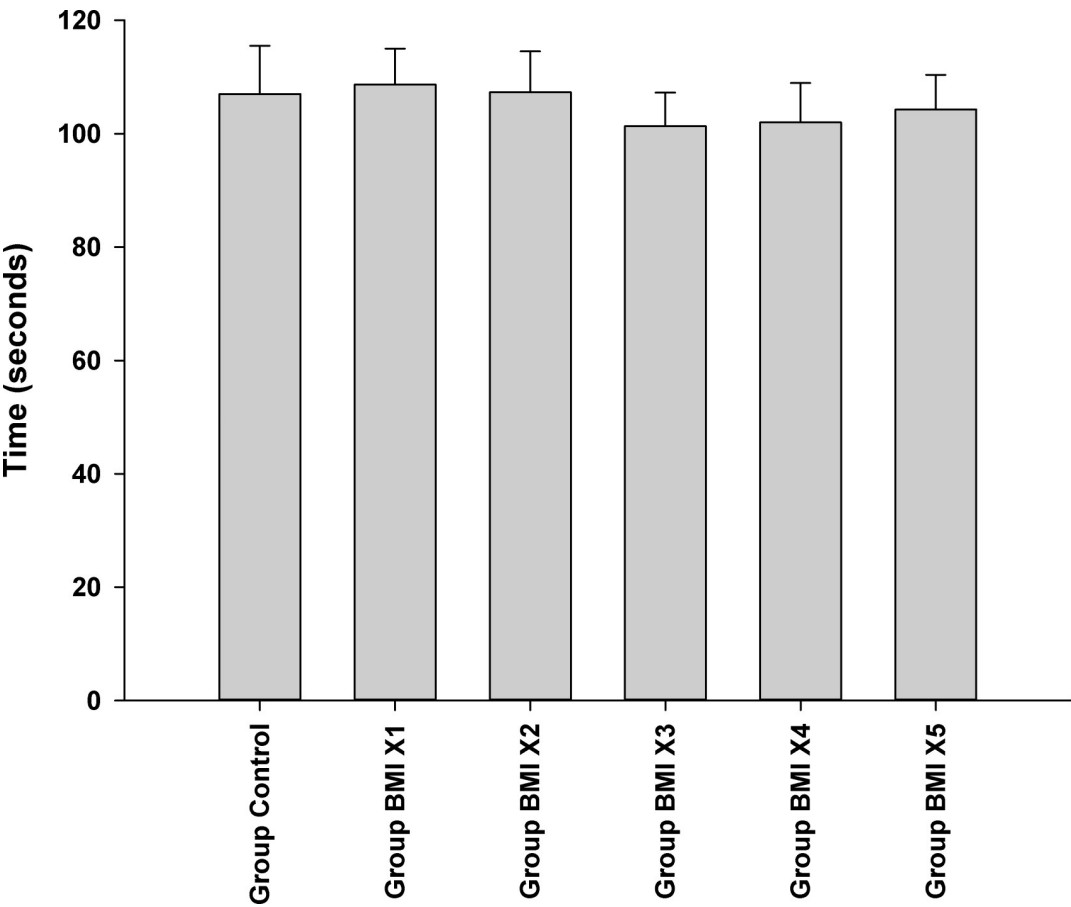

**Fig 6. Motor function tests.** The graphs are presented as means ± standard deviation of the time at which the rat fell off the Rota-Rod.

BMI1008 is a newly developed experimental drug that is composed of lidocaine, MB, vitamin B, and dexamethasone. Lidocaine is a widely used local anesthetic [15], and the other components of BMI1008 except lidocaine (MB, vitamin B, and dexamethasone) have been used as adjuvants of local anesthetics to reduce pain.

Local anesthetics are widely used in clinical practice for regional anesthesia or analgesia via central neuraxial blockade, peripheral nerve block, or local infiltration. Local infiltration around the nerve can produce analgesia by interrupting the conduction of pain signals to the brain [16]. However, the duration of the action of local anesthetics limits these benefits because the analgesic effect of a nerve block lasts only a few hours. Further, cardiovascular and central nerve system toxicity are very important problems when using local anesthetics at high doses [17]. Thus, adjuvant drugs, such as opioids, vasoactive agents, alpha2-adrenergic agonists, steroids, and nonsteroidal anti-inflammatory drugs, have been applied to decrease the toxicity of the local anesthetics while increasing the quality and duration of anesthesia and analgesia [18, 19]. In the present study, higher dose of BMI-L with constant 1% lidocaine showed prolonged analgesic effect in a dose-dependent manner. Thus, BMI-L could be suggested as a possible adjuvant to local anesthetics.

MB, one of the BMI1008 components, is a low-molecular-weight dye. It has been used in many different clinical fields, including anorectal surgery and discogenic pain intervention, and has been reported to reduce postoperative pain [20, 21]. The suggested mechanisms for

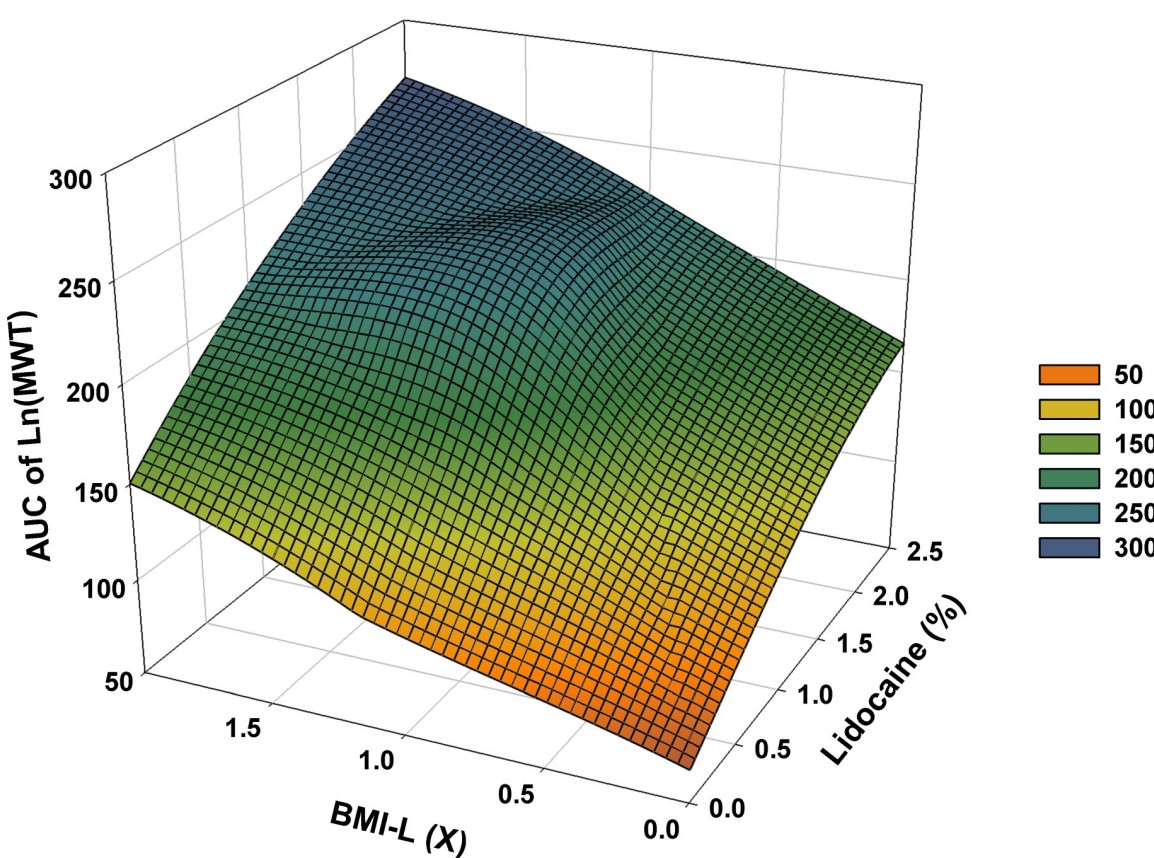

**Fig 7. Response surface for combination of lidocaine and BMI-L.** The surface plots were constructed based on the individual dose of lidocaine and BMI-L and the AUC of Ln(MWT) for 3 d for each dose. AUC: area under curve, BMI-L: other components of BMI1008 except lidocaine, Ln: natural log, MWT: mechanical withdrawal threshold, X: times.

the analgesic effect of MB are inhibition of voltage-gated sodium channels (VGSCs) [22], inhibition of nitric oxide (NO) synthase [23], inhibition of monoamine oxidase (MAO) [24], inhibition of free-radical generation [25], destruction of free nociceptive nerve endings [26], and neuroprotective effectiveness [27]. Among those, the neurotropic effects of MB, such as destruction of free nociceptive nerve endings and blocking nerve conduction, were suggested as mechanisms for its analgesic effect, which facilitates direct MB injection at the painful sites [20]. Moreover, the anti-inflammatory effect from the inhibition of NO synthase also contributes to the analgesic effect of MB. Although a dose-dependent analgesic effect was reported when MB was administered systemically [28], the dose-dependent analgesic effect of the local application of MB, which is commonly used in clinical practice, is not reported. The present study demonstrates the dose-dependent analgesic effects of locally administered MB. In clinical studies, the effect of MB showed conflicting results between studies [29, 30]. Referring to the present study, it is thought that increasing the MB concentration or additional adjunctives are needed to obtain significance for the effect of MB in clinical practice.

Dexamethasone, another component of BMI1008, is a glucocorticoid that has a number of beneficial properties in the perioperative period. It has antiemetic, anti-inflammatory, antipyretic, and anti-allergic properties [6]. Dexamethasone has also been reported to prolong the analgesic duration of local anesthetics in several clinical fields, including brachial plexus blockades, intercostal nerve blocks, and intravenous regional anesthesia [31]. Although the

effectiveness of dexamethasone alone was not directly compared, our study showed that an increased dose of BMI-L increased the analgesic effect and its duration. In our study, BMI1008 increased MWT compared with fentanyl at postoperative 6 and 8 h. In the study of Hval et al., dexamathasone delayed the onset of the analgesic effect but prolonged the analgesic effect up to 3 d AS [32]. We assume that the property of dexamethasone that prolongs the duration of the analgesic effect affected our study results. However, the plasma elimination half-life of dexamethasone is only about 6 h [33], suggesting a persistent drug effect unrelated to its plasma concentration. Because glucocorticoids inhibit transcription [34], changes in protein expression can be expected to persist after the drug is cleared from the plasma. Dexamethasone not only prolonged the duration of anesthesia but also reduced postoperative pain in several studies [35]. The mechanism of glucocorticoids is reducing prostaglandin synthesis by inhibiting both phospholipase enzyme and cyclooxygenase Type II and by decreasing the products of cyclooxygenase-2 [36]. They also modulate the inflammatory response by inhibiting tumor necrosis factor-α, interleukin 1β, interleukin 6, c-reactive protein, and leukocyte receptors [37]. The analgesic effect of glucocorticoids has been demonstrated in a variety of surgical procedures [32].

The vitamin B family is known to play a very important role in nerve conduction and excitability [38], and their deficiencies are associated with pain disorders [39]. Therefore, vitamin B has been used in some painful disorders such as polyneuropathy, neuralgia, radiculopathy, neuritis associated with pain paresthesia, and diabetic peripheral neuropathy [40]. In animal studies, vitamins B1 (thiamine), B6 (pyridoxine), and B12 (cyanocobalamin) and their combination inhibit thermal- and chemical-induced pain [41]. In addition, several studies have demonstrated that vitamin B alleviates neuropathic pain involving acute pain [38]. Thus, the vitamin B family is given with other analgesics for acute pain, such as postoperative pain [42]. In our study, higher doses of BMI-L showed greater analgesic effects. This result also suggests that MB, dexamethasone, and vitamin B have an additive analgesic effect on the analgesic effect of lidocaine. No case for neurotoxicity was found in our study. However, possibilities for neurotoxicity from combining more than one drug cannot be ruled out entirely.

There are some limitations of our study. First, in our study, we only assessed mechanical hyperalgesia in the investigation of the analgesic effect of BMI1008. Heat and cold allodynia was not investigated. Thus, further investigation is needed to describe the entire analgesic profile of BMI1008. Second, the behavioral assessment was measured frequently; thus, full recovery from an earlier test may not have been achieved prior to a follow-up test. Although there was at least a 15-min interval between each MWT assessment, an earlier MWT measurement could still have affected the outcome of a subsequent MWT measurement. However, the time between the von Frey filament tests was > 10 min and was enough for rats to adapt to the environment. Third, since the author did not reveal the mechanism of each component of BMI1008 using a selective receptor antagonist, this study could not suggest an overlapping drug mechanism when using a multimodal strategy. Last, this study showed that the analgesic effect was better as the concentration of BMI1008 increased and that the 5-fold concentrate of BMI1008 (lidocaine 40 mg/kg) had no effect on the motor function. Despite there are other studies conducted with high-dose lidocaine in rat experiments [43], high concentrated BMI1008 would be difficult to use to human in actual clinical practice because of toxicity. Therefore, it is not possible to suggest the optimal dose to be introduced directly into clinical practice through this study.

In conclusion, BMI1008 showed its analgesic effect in a rat model of incisional pain in a concentration-dependent manner. Moreover, other components of BMI1008 except lidocaine showed an additive effect on the analgesic effect of lidocaine.

## Supporting information

**S1 File. Data for Experiment 1: Evaluation of concentration-dependent analgesic effect of BMI1008.**
(XLSX)

## Author Contributions

**Conceptualization:** Geun Joo Choi, Eun Jin Ahn, Oh Haeng Lee, Hyun Kang.

**Data curation:** Geun Joo Choi, Eun Jin Ahn, Oh Haeng Lee, Hyun Kang.

**Formal analysis:** Geun Joo Choi, Hyun Kang.

**Funding acquisition:** Geun Joo Choi, Hyun Kang.

**Investigation:** Eun Jin Ahn, Oh Haeng Lee.

**Methodology:** Geun Joo Choi, Eun Jin Ahn, Oh Haeng Lee.

**Project administration:** Oh Haeng Lee.

**Resources:** Eun Jin Ahn.

**Supervision:** Hyun Kang.

**Writing – original draft:** Geun Joo Choi, Eun Jin Ahn, Oh Haeng Lee.

**Writing – review & editing:** Hyun Kang.

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
