## [Decision Letter · Decision Letter 0]

30 Jun 2021

PONE-D-21-13470

Effects of a BMI1008 Mixture on Postoperative Pain in a Rat Model of Incisional Pain

PLOS ONE

Dear Dr. Kang,

Thank you for submitting your manuscript to PLOS ONE. After careful consideration, we feel that it has merit but does not fully meet PLOS ONE’s publication criteria as it currently stands. Therefore, we invite you to submit a revised version of the manuscript that addresses the points raised during the review process.

We look forward to receiving your revised manuscript.

Kind regards,

Prof. Dr. Dragan Hrnčić, MD, MSc, PhD

Academic Editor

PLOS ONE

Journal Requirements:

2. 'To comply with PLOS ONE submissions requirements, please provide methods of sacrifice in the Methods section of your manuscript.

"HK:

BMI Korea supported BMI 1008 materials and research grants for this research.

No role in the study design, data collection and analysis, decision to publish, or

preparation of the manuscript.

GJC:

This research was supported by the Basic Science Research Program through the

National Research Foundation of Korea (NRF) funded by the Ministry of Education,

Science and Technology [Grant No. NRF-2020R1C1C1011263].

No role in the study design, data collection and analysis, decision to publish, or

preparation of the manuscript"

We note that you received funding from a commercial source: "BMI Korea"

Reviewers' comments:

Reviewer's Responses to Questions

**Comments to the Author**

1. Is the manuscript technically sound, and do the data support the conclusions?

Reviewer #1: Yes

Reviewer #2: Yes

Reviewer #3: Partly

2. Has the statistical analysis been performed appropriately and rigorously? 

Reviewer #1: Yes

Reviewer #2: Yes

Reviewer #3: Yes

3. Have the authors made all data underlying the findings in their manuscript fully available?

Reviewer #1: Yes

Reviewer #2: Yes

Reviewer #3: Yes

4. Is the manuscript presented in an intelligible fashion and written in standard English?

Reviewer #1: Yes

Reviewer #2: Yes

Reviewer #3: No

5. Review Comments to the Author

Reviewer #1: Abstract

Background

-Explain what is BMI1008 in the background of the abstract?

Introduction

-Dexamethasone is a glucocorticoid that has been used to reduce inflammation and tissue damage in a variety of conditions, including inflammatory bowel disease, rheumatoid arthritis, and some malignancies, and its analgesic and anti-inflammatory effects after surgery have been explored. – Why did you mention inflammatory bowel disease? Is it only inflammatory state for the use of glucocorticoids? The point of this sentence should be the mechanism of analgesic action and effect of glucocorticoids?

-Vitamin B complex and 5 85 NaHCO3 have also been extensively studied for their pain-modulating effects. – How?

Methodology

Motor function tests

-There were only three rats per group in motor function examination. Why? In all other experiments there were 10-12 rats what is good for statistical significance.

We also drew the response surface analysis plot of the drug combination to summate the analgesic effect of lidocaine and BMI-L. The analgesic effect of each combination was calculated using means of the area under the curve (AUC) for 3 d in Ln(MWT). Moreover, seven points of drug combination were selected when BMI-L was less than 2x and the concentration of lidocaine was less than 2.5%. –

-Reference?

-Why did you choose these method of examination of the summary of the drugs?

Results

-It will be more appropriate to show the results of motor function test through graph.

Response surface analysis

The AUC of Ln(MWT) showed an approximately linear positive correlation with BMI-L and lidocaine concentration. Moreover, co-administration of BMI-L and lidocaine augmented the AUC of Ln(MWT) compared with the administration of single drugs. Gross concavity of the plane was not observed, suggesting a higher possibility of the additivity assumption and a lower possibility of the antagonism assumption (Figure 6).

- How did you conclude this? Is it based on some previous work? Why did you choose this method? There are a lot of others more concrete methods for the interactions of the drugs. Maybe, it would be better to add some other method for this part of results.

Discussion

The additivity of lidocaine and BMI-L appeared more likely than antagonism when these drugs were co-administered.

-The same remark as I mentioned in the results.

Reviewer #2: Paper titled (Effects of a BMI1008 Mixture on Postoperative Pain in a Rat Model of Incisional Pain)

- Generally, the authors did very hard work,I read it more time but it is confusing so if they try to simplify the measurements

and the result

- Is the BMI1008 combination is approved or registered

- the combination of more than one drugs may result in neurotoxicity

- Is vitamin b has local effect on nerve cells or axone

Reviewer #3: This is an experimental study to confirm the control of incisional pain according to the dose of BMI1008 and BMI-L. However, authors need to correct the manuscript overall.

MP4, L84-85 . Please add a reference.

MP18, L390 "The additivity of lidocaine and BMI-L appeared more likely than

antagonism when these drugs were co-administered." Can this dose-response relationship be explained as antagonism? I cannot understand. Please explain more detail.

The discussion section is not organized in general. Rearrange all paragraphs. For example, you need not to elaborate on the( MP18, L396-399) incisional pain model. Rather, it would be better to find the literature on the dose-response relationship of drugs and add them. Or please add more details such as analgesic effect when used in actual anesthesia clinical practice

MP19 L410 It has been reported that vasoactive agents and alpha2-adrenergic agents increase the quality and duration of local anesthetics. For example, which drugs do you mean? Are there any drugs other than epinephrine that have such an effect?

MP21 464 Is it correct to use the word "hyperalgesia"? Do you mean "mechanical stimulation"?

It is very important to write in the correct terminology.

I have a question, are BMI1008 and BMI-L drugs that the authors prepared and used in the study? Is it a drug marketed under a brand name?

6. PLOS authors have the option to publish the peer review history of their article (what does this mean?). If published, this will include your full peer review and any attached files.

Reviewer #1: No

Reviewer #2: No

Reviewer #3: No

---

## [Author Response · Author response to Decision Letter 0]

13 Aug 2021

We attached the "response to reviewers" file as we have some figures in it.

---

## [Decision Letter · Decision Letter 1]

31 Aug 2021

Effects of a BMI1008 Mixture on Postoperative Pain in a Rat Model of Incisional Pain

PONE-D-21-13470R1

Dear Dr. Kang,

We’re pleased to inform you that your manuscript has been judged scientifically suitable for publication and will be formally accepted for publication once it meets all outstanding technical requirements.

Kind regards,

Prof. Dr. Dragan Hrncic, MD, MSc, PhD

Academic Editor

PLOS ONE

Additional Editor Comments (optional):

Reviewers' comments:

Reviewer's Responses to Questions

**Comments to the Author**

1. If the authors have adequately addressed your comments raised in a previous round of review and you feel that this manuscript is now acceptable for publication, you may indicate that here to bypass the “Comments to the Author” section, enter your conflict of interest statement in the “Confidential to Editor” section, and submit your "Accept" recommendation.

Reviewer #1: All comments have been addressed

Reviewer #2: All comments have been addressed

Reviewer #3: All comments have been addressed

2. Is the manuscript technically sound, and do the data support the conclusions?

Reviewer #1: Yes

Reviewer #2: Yes

Reviewer #3: Partly

3. Has the statistical analysis been performed appropriately and rigorously? 

Reviewer #1: Yes

Reviewer #2: Yes

Reviewer #3: Yes

4. Have the authors made all data underlying the findings in their manuscript fully available?

Reviewer #1: Yes

Reviewer #2: Yes

Reviewer #3: Yes

5. Is the manuscript presented in an intelligible fashion and written in standard English?

Reviewer #1: Yes

Reviewer #2: Yes

Reviewer #3: Yes

6. Review Comments to the Author

Reviewer #1: Authors improved their work. They accepted all suggestions and explained all questions I had asked. I suggest to accept manuscript for the publication.

Reviewer #2: (No Response)

Reviewer #3: (No Response)

7. PLOS authors have the option to publish the peer review history of their article (what does this mean?). If published, this will include your full peer review and any attached files.

Reviewer #1: No

Reviewer #2: No

Reviewer #3: No

---

## [Editor Report · Acceptance letter]

7 Sep 2021

PONE-D-21-13470R1 

Effects of a BMI1008 Mixture on Postoperative Pain in a Rat Model of Incisional Pain 

Dear Dr. Kang:

I'm pleased to inform you that your manuscript has been deemed suitable for publication in PLOS ONE. Congratulations! Your manuscript is now with our production department. 

Kind regards, 

on behalf of

Professor Dragan Hrncic 

Academic Editor

PLOS ONE